# Construction of Wearable Touch Sensors by Mimicking the Properties of Materials and Structures in Nature

**DOI:** 10.3390/biomimetics8040372

**Published:** 2023-08-17

**Authors:** Baojun Geng, Henglin Zeng, Hua Luo, Xiaodong Wu

**Affiliations:** School of Mechanical Engineering, Sichuan University, Chengdu 610065, China

**Keywords:** flexible touch sensor, biomimetics, bio-inspired material, bio-inspired structure, bio-inspired sensor

## Abstract

Wearable touch sensors, which can convert force or pressure signals into quantitative electronic signals, have emerged as essential smart sensing devices and play an important role in various cutting-edge fields, including wearable health monitoring, soft robots, electronic skin, artificial prosthetics, AR/VR, and the Internet of Things. Flexible touch sensors have made significant advancements, while the construction of novel touch sensors by mimicking the unique properties of biological materials and biogenetic structures always remains a hot research topic and significant technological pathway. This review provides a comprehensive summary of the research status of wearable touch sensors constructed by imitating the material and structural characteristics in nature and summarizes the scientific challenges and development tendencies of this aspect. First, the research status for constructing flexible touch sensors based on biomimetic materials is summarized, including hydrogel materials, self-healing materials, and other bio-inspired or biomimetic materials with extraordinary properties. Then, the design and fabrication of flexible touch sensors based on bionic structures for performance enhancement are fully discussed. These bionic structures include special structures in plants, special structures in insects/animals, and special structures in the human body. Moreover, a summary of the current issues and future prospects for developing wearable sensors based on bio-inspired materials and structures is discussed.

## 1. Introduction

Wearable and flexible touch sensors are electrical devices that can translate pressure or force into electronic signals with a wide range of applications, including electronic skins [1], soft robotics [2], and human–computer interaction [3,4,5]. In addition, wearable touch sensors can be used to detect the vital signs (e.g., pulse signal, heart rate, respiration, speech, joint movements, etc.) related to human health conditions [6,7,8,9,10,11,12], making them indispensable for wearable health monitoring and personal healthcare in the future [13,14]. Flexible touch sensors have made considerable advancements in recent years, reporting on ultrahigh device performance (e.g., high sensitivity, broad working range, fast response, low detection limit, low power consumption, etc.) [9,15,16,17,18] and diverse functionalities (e.g., biocompatibility, self-powering capability, biodegradability, self-healing capability, etc.) [19,20,21,22]. In the construction of the aforementioned wearable touch sensors, a variety of novel materials (e.g., flexible polymers [23], stretchable elastomers [24], soft hydrogels [25], etc.) and unique structures (e.g., pyramid [26], hemisphere [27], cylinder [28], cilia [29], crack [30,31], etc.) have played an importance role in the enhancement and realization of high sensing performance and unique device functionalities.

Creatures in nature (including plants, insects, animals, etc.) have evolved over millions of years, developing a diversity of special skills and adaptive capacities. In the evolutionary process of various creatures, the unique properties of biomaterials and biological structures play a crucial role. These biogenetic materials and structures possess outstanding properties derived from natural selection. These properties include good flexibility or even stretchability, biological self-healing capability, high sensitivity to external stimulations (e.g., mechanical and thermal stimuli [32]), mechanoluminescence properties [33], and multifunctionality enabled by hierarchical structures [34]. The development of artificial touch sensors is always inspired by the unique characteristics of biomaterials and biological structures.

By mimicking the unique properties (e.g., softness, stretchability, ionic conduction, self-healing, special surficial patterns, hierarchical structures, etc.) of natural materials and structures, a novel range of wearable touch sensors with extraordinary characteristics can be developed. For instance, ionically conducting hydrogel and self-healing materials can be utilized to fabricate bio-inspired touch sensors and artificial e-skins with good softness, self-repairing capability, and skin-attaching comfort [21]. In addition, biological hierarchical structural can be mimicked to construct novel touch-sensing devices with both mechanical sensitivity and a broad detection range [35]. Therefore, mimicking natural materials and structures holds promise for developing sustainable, high-performance, and biocompatible wearable touch sensors. Bio-inspired flexible touch sensors also exhibit tremendous application potential in various fields, particularly for intelligent soft robots and wearable health monitoring.

Related reviews of wearable and flexible touch sensors have been conducted [13,36,37], providing valuable references and guidance for future research. However, most review papers mainly focus on materials innovations, fabrication methodology, performance improvement, and potential applications, rarely emphasizing the significance of the biomimetic approach in designing and fabricating novel touch sensors with enhanced performance. In this review, we outline the current research status of bio-inspired wearable touch sensors by mimicking the properties of materials (e.g., hydrogel materials, self-healing materials, and other bio-inspired materials) and structures (including structures of plants, animals, and humans) found in nature (Figure 1). We categorize our paper into two implementations: the fabrication of flexible touch sensors with unique features that exist in natural materials and the enhancement of touch sensor performance by employing biological structures. Furthermore, we summarize the existing challenges of bionic touch sensors and present an outlook for the development of wearable touch sensors.

## 2. Bio-Inspired Materials for the Construction of Flexible Touch Sensors

Living creatures that survive in the long process of natural selection exhibit extraordinary abilities and functionalities to adapt to changes in the environment. Many of these special abilities and functionalities arise from natural materials with unique biological, mechanical, or even electrical properties (e.g., softness, toughness, stretchability, self-healing ability, mechanoluminescence, etc.) [56,57,58]. Scientists are always seeking and creating artificial materials to mimic these remarkable properties, which is especially frequently common in the design and construction of novel touch sensors using biological and bio-inspired materials. In this section, we will summarize the current developments in the fabrication of novel flexible touch sensors based on bio-inspired materials, including hydrogels, self-healing materials, and other kinds of biomimicking materials (Table 1).

### 2.1. Touch Sensors Based on Hydrogel Materials

Hydrogels are 3D cross-linked hydrated polymer networks that, like human organs, can store exceptionally high quantities of water. The mechanical properties of the hydrogels, such as their stretchability, toughness, flowability, and softness, can be freely altered and controlled during the synthesis process [67,68]. To gain electrical or ionic conductivity of hydrogels for sensing functionality, conducting fillers or ionic salts can be introduced into the hydrogels [69]. Additionally, the mechanical properties, electrochemical characteristics, and biological functions of creatures can be simulated by modifying the network structures or functional groups of hydrogels [68,70], providing great possibilities for designing and fabricating flexible touch sensors. Hydrogel materials, similar to human tissue, contain many water molecules. Unlike rubber and plastic materials, hydrogels exhibit exceptional mechanical flexibility and softness, resembling human skin. Additionally, hydrogels possess favorable features such as transparency and shape controllability, and have proven to hold great promise for the fabrication of wearable touch screens and other light-transparent human–machine interaction devices [59,61,71]. Furthermore, hydrogel materials exhibit excellent biocompatibility, rendering them harmless to the human body. This characteristic makes hydrogels ideal candidates for the development of wearable sensors capable of long-term monitoring of human motion signals [72,73]. Moreover, hydrogel-based touch sensors with outstanding self-adhesion and high toughness have been developed [74,75]. This functionality obviates the need for additional adhesive tape when affixing the sensors to the skin, facilitating the detection of subtle mechanical signals from the skin surface. Hydrogel materials can also have unique features and different functionalities by changing the conductive filler, dopant, cross-linker, or hydration state, exhibiting great application potential in the emerging areas of bionic flexible devices [76], flexible energy storage devices [77], and human–machine interfaces [78].

Hydrogels have outstanding softness, flexibility, stretchability, high transparency, and biocompatibility, making them ideal candidates for constructing electrodes in flexible sensors. Sarwar et al. developed a new kind of touch sensor using stretchy conductive hydrogel as the electrodes [71]. This flexible sensor overcomes the trade-off between optical transparency and electrical conductivity. The conductive hydrogel remains highly transparent due to its dielectric properties at optical wavelengths. Although the electrical conductivity of these hydrogel electrodes is lower than that of indium tin oxide by a factor of 1000, it does not hinder their capacitive sensing performance. By coupling these ion–water gel electrodes with silicone-based elastomer capacitors, a novel ion skin can be created. It exhibits strong interaction with proximal fingers, allowing for easy differentiation between finger touch and stretching and bending. This technology enables the operating mode of flexible sensors to extend to close-range detection of fingers.

Additionally, hydrogels can also serve as desirable electrode materials for fabricating triboelectric nanogenerator-based touch-sensing devices. Pu et al. created a soft, skin-like triboelectric nanogenerator (STENG) that utilizes ionic hydrogels as electrodes to harvest biomechanical energy and realize touch-sensing functionality (Figure 2a) [59]. These hydrogel electrodes possess skin-like mechanical properties and can be attached obediently to the skin, with high signal accuracy and exceptional wearing comfort. The STENG consists of two elastomeric membranes, polydimethylsiloxane (PDMS) or 3M VHB 9469, which enclose the ionic hydrogels (PAAm-LiCl hydrogels) composed of polyacrylamide (PAAm) hydrogels and lithium chloride (LiCl). Electrical connections are established by attaching Al strips or Cu wires to the hydrogel electrodes. This STENG, with high transparency to visible colors (Figure 2c) and ultrahigh stretchability (Figure 2d), can operate in a single electrode mode sensing modality (Figure 2b) and can achieve instantaneous surface power densities of 35 mW m^−2^ and open-circuit output voltages of up to 145 V. The electronic skin made from the hydrogel-based STENG exhibits a mechanical sensitivity of 0.013 kPa^−1^ (Figure 2e) and a detection limit of 1.3 kPa (Figure 2f), making it a desirable soft touch sensor that can conformally attach to the dynamically curved skin (Figure 2g).

Recently, Sun developed an “ionic skin” that uses hydrogels or other ionic gels as conductors for ionic conduction. This strategy greatly enhances the designability of artificially intelligent skins with touch-sensing functionality that combine biocompatibility and high stretchability [79]. In addition, Lei and associates have created an ionic skin using a bio-inspired supramolecular mineral hydrogel [60]. This hydrogel is composed of amorphous calcium carbonate (ACC) nanoparticles, physically cross-linked alginate, and polyacrylic acid (PAA) chains (Figure 2h). The resulting ionic skin is self-healable, highly sensitive, and mechanically flexible. The ACC/PAA/alginate hydrogel is formable, elastic, and highly stretchable (Figure 2j) and can conform well to curved or dynamic surfaces, (such as a dynamic prosthetic finger Figure 2k). This is because Ca^2+^ in alginate possesses a stronger chelating effect than PAA. By introducing an appropriate ratio of alginate, the hydrogel can be put in an excellent semi-solid state, i.e., when the material storage modulus is essentially equivalent to the loss modulus (Figure 2i). Using the developed hydrogel, a capacitive pressure sensor was created by combining two hydrogel films with a dielectric layer, as shown in Figure 2l [60]. The change in capacitance of the sensor is in a good linear relationship with the compressive pressure (Figure 2m). Furthermore, the pressure sensitivity of this hydrogel material (with a calculated sensitivity of 0.17 kPa^−1^) is superior to the sensors based on conventional PAAm hydrogels [79], and other materials.

To overcome the issues related to rigid electrodes and complex electrode configuration in the fabrication of current touch panels, Kim et al. demonstrated a hydrogel-based ionic touch panel [61]. As an ionic conductor, a strip of PAAm hydrogel including LiCl is attached to platinum electrodes on both sides, forming a one-dimensional (1D) ionic touch strip (Figure 2n). When a finger contacts the hydrogel strip, current flows from both ends of the strip toward the touch position. The currents I_1_ and I_2_ were measured by ammeters A_1_ and A_2_, respectively (Figure 2o). With the touch point moving from left to right, I_1_ decreases linearly while I_2_ increases linearly (Figure 2p); however, their sum remains constant. Notably, stretching the gel strip (Figure 2p(II)) leads to an expansion in the strip’s area, increasing its parasitic capacitance. Consequently, both the touch-induced current and the baseline current will rise compared to the non-stretched states. Based on the sensing principle of a 1D touch strip, a 2D hydrogel touch panel (Figure 2q) with a 2D surface capacitive touch-sensing capability was constructed. Figure 2r depicts the schematic design of the touch panel, which is attached to the arm by a VHB film with a 1 mm thickness. The hydrogel material is completely transparent, with a 98% transmission of visible light, and can be operated at over 1000% strain. The subject can comfortably wear this touch panel for various human–machine interfacing tasks, such as writing text (Figure 2s) and playing piano (Figure 2t).

Most hydrogel materials suffer from dehydration problems. To eliminate this issue, Lei et al. report a new adaptive polyionic elastomer via a rational molecular design [62]. The polyionic elastomers are obtained by one-step polymerization of the cationic monomer DMAEA-Q (methyl chloride quaternized N,N-dimethylaminoethyl acrylate) in the presence of linear polyanionic PAA. Such polyionic elastomers possess unique advantages (such as autonomous self-healing, and 3D printing capabilities) when compared to stretchable silicone [61], self-healing elastomers [65], and ionic conductive hydrogels [80]. Based on this polyionic elastomer, a polyionic skin with an integrated iontronic sensor was designed and fabricated based on 3D printing technology (Figure 2u). This polyionic skin possesses various characteristics (such as transparency, mechanical flexibility, and self-adherent ability), and importantly, it exhibits better mechanical adaptation than other electronic skin in the detection of strain, pressure, touch, humidity, and temperature (Figure 2v,w).

### 2.2. Touch Sensors Based on Self-Healing Materials

Human skin has an impressive ability to restore its diverse sensing functionalities thanks to its self-repairing capabilities [80,81]. In recent years, researchers aim to create bionic electronic skin with similar self-healing characteristics. A hot research topic is to find skin-like materials with reproducible self-healing capabilities, as well as mechanical and thermal sensing abilities. One technical approach to creating self-healing sensors is to combine self-healing polymer substrates with functionalized inorganic nanomaterials. For example, polyurethane can be mixed with MXene to achieve excellent self-healing and conducting capabilities [50]. Alternatively, researchers are also exploring new self-healing mechanisms in flexible materials to improve the recovery efficiency of flexible touch sensors. For instance, a novel type of self-healing thermoplastic elastomer (PBPUU) for underwater use was created by Khatib and colleagues using terminal hydroxy polybutadiene (HTPB), isophorone diisocyanate (IPDI), and 4-aminophenyl disulfide (APDS) [82].

Many flexible conductive composite materials can be used to fabricate touch sensors, but it is challenging to incorporate self-healing functions into these materials and sensors. Tee, B. C. et al. demonstrated a reproducible, room-temperature self-healing electronic sensor skin made from a composite material consisting of supramolecular organic polymers and embedded nickel nanostructured particles [83]. The composite is composed of a supramolecular polymer hydrogen-bonded network with a glass transition temperature (T_g_) below room temperature. The composite contains micro-nickel (Ni) particles that are chemically compatible with the polymer (Figure 3a). The composite offers good mechanical flexibility and tunable electrical conductivity, enabling the initial conductivity to be recovered with over 90% efficiency after rupturing. The mechanical properties can be fully restored after approximately 10 min, and the sensors can reliably detect the changes in pressure and limb positions (Figure 3b,c).

In addition to conductive composite materials, Oh et al. developed a new kind of semiconducting material with room-temperature self-healing capability and mechanical stretchability [63]. For the fabrication of the semiconducting material, poly(3, 6-di(thiophen-2-yl) diketopyrrolo [3, 4-c] pyrrole-1, 4-dione-alt-1, 2-dithienylethene) with 10 mol% 2, 6-pyridinedicarboxamine moieties (DPP-TVT-PDCA) was employed because of its good charge carrier mobility. PDCA units were incorporated into the semiconducting material, which can bind well to the insulating and stretchable poly(dimethylsiloxane-alt-2,6-pyridinedicarbozamine) (PDMS-PDCA) polymer. The self-repairing capability of this material originates from the dynamic cross-linking of PDMS and DPP through Fe (III)-PDCA complexation. Moreover, the Fe (III)-PDCA coordination has multiple dynamic bonds of dissimilar strengths, which promotes the intrinsic tensile and self-repairing ability of dynamic cross-linking (Figure 3d). The sensing capability of the active-matrix transistor sensor arrays fabricated using this novel self-healing material was verified using a finite-element method. The method was employed to evaluate the distribution of applied strain with a plastic tip (Figure 3e–g).

In addition to rigid self-healing touch sensors, by enclosing three-dimensional electrodes in a self-healing foam material, Guo et al. suggested a low-modulus biomimetic artificially innervated porous foam. [64]. To mimic the somatosensory innervation system in human skin, they used 3D wire electrodes as the “nerves”. The foam material is synthesized by a one-step self-foaming process and comprises a low-modulus elastomer composed of cross-linked polymer chains with 1,3-diaminopropane (DAP) and surfactant molecules, along with micro-nickel particles [54]. The surfactant can be trapped in the polymer matrix by the dipole-dipole interactions between the polymer chains and the surfactant molecules, which enables the foam material to self-heal under mild heating conditions (Figure 3h). Modulation of the conductive metal particles’ concentration in the material allows the construction of a novel kind of touch sensor with both capacitive and piezoresistive sensing modalities.

To extend the application scenarios in both dry and wet environments, Cao et al. presented a transparent, skin-like material inspired by jellyfish which can repair itself autonomously regardless of dry and wet conditions [65]. This contrasts with previous approaches using hydrogen bonding and metal-ligand coordination to create self-healing materials (Figure 3i). This material contains a stretchable fluorocarbon elastomer with a high chain dipole moment and fluorine-rich ionic liquids called GLASSES. Due to its reversible ion–dipole interaction, the material provides rapid and reproducible self-repairing capability in acidic or alkaline environments. The resultant material also exhibits an ionic conductivity of up to 10^−3^ S cm^−1^ and can withstand up to 2000% of strain. A conformal pressure sensor was created by placing a piece of GLASSES on top of a 3D-printed “moon” surface. The LED light’s brightness fluctuates between distinct touch zones and its intensity changes in response to various touch stimulations (Figure 3j), demonstrating prospective using in water-based soft robots and waterproof human–machine interfaces.

### 2.3. Touch Sensors Based on Other Bionic Materials

In addition to hydrogels and self-healing materials, other bionic materials with intriguing properties or functionalities could be employed to construct novel flexible touch sensors. For example, materials with ionic channel-mimicking properties can be used to construct highly sensitive touch sensors [41]. Natural human skin can be used as the sensing layer for fabricating sensitive capacitive touch sensors [66]. Light-emitting materials can be used to mimic the color-changing behaviors of chameleons [33,46]. In addition, bio-inspired temperature-sensing films with thermally regulated ionic mobility can be used to create biomimetic touch sensors [44]. These aforementioned bio-inspired materials greatly expand the scope and application scenarios of wearable touch sensors.

Biological materials have continuously been a source of innovation for researchers in developing synthetic materials or devices that can mimic the unique functions of biological cells. Recently, Amoli et al. proposed synthetic multicellular hybrid ion pump (SMHIP) materials inspired by the properties of biological multicellular structures [41]. The material was designed to imitate the ion pump behaviors using 1-ethyl-3-methy-limidazoliumbis (trifluoromethyl-sulfonyl) imide ([EMIM^+^] [TFSI^−^] ion pairs) that are assembled in situ on the surface of silica microspheres. In the proposed synthetic multicellular membrane, the [EMIM^+^] [TFSI^−^] ion pairs were bound to the surface of the silica microstructure that was embedded in the TPU matrix (Figure 4a). The SMHIP films feature invertible pumping of ions under external stimuli, and a double electric layer (EDL) was observed at the IL-SiO2-TPU/electrode interface (Figure 4b,c). The reversible movement of ions was primarily due to the breakage and recurrence of [TFSI^−^]-silica H-bonds and the [EMIM^+^] cation’s π-π stacking interactions. Finally, an ion mechanoreceptor skin was prepared using this SMHIP material. Such ion mechanoreceptor skin is highly sensitive (5.77–48.1 kPa^−1^) over a broad range of pressures (0–135 kPa). The mechanical sensitivity of this bio-inspired mechanoreceptor even surpasses the pressure-sensing ability of natural skin mechanoreceptors, such as Merkel cells and Meissner’s corpuscles [84].

In addition to synthesizing artificial materials, nature materials (e.g., human skin, etc.) can be directly used to fabricate touch sensors. As an example, Zhu and colleagues have developed a skin–electrode mechanosensing structure (SEMS) that uses the ion transport properties in living systems, like human skin tissue [66]. The SEMS sensor consists of a sensing electrode (SE) with a microstructured surface and a compliant counter electrode (CE) (Figure 4d). Applying pressure to the SE changes the contact area between the micro-pillar structure and the skin surface, thus creating a capacitor between CE and SE with a much higher capacitance change than traditional capacitor-based touch sensors (Figure 4e). The SEMS sensor has a simpler structure than conventional touch sensors and does not require synthetic ionic gels or hydrogels. Additionally, a full fabric SEMS-based smart glove with good pressure mapping capability at millimeter spatial resolution was demonstrated (Figure 4f). The glove can be used to grip a soft compressible balloon, producing pressure distribution that is adequately uniform throughout the palm. (Figure 4g). In addition, holding a hard microphone with the glove produces a strong signal amplitude on the fingers and a weak signal on the palm (Figure 4h). The stability and simplicity of this SEMS sensor delivers a promising future for healthcare and human–machine interaction applications.

The allochroic behaviors of chameleons also inspired the design and fabrication of optical flexible touch sensors that convert mechanical force in the color or brightness changes. The piezoelectric photonic effect, a bidirectional coupling effect between piezoelectric and photoexcitation properties, could be used to construct light-emitting sensors and electronic skins [85]. The piezoelectric photonic effect can cause a mechanoluminescence (ML) process, which converts mechanical stress into visible light emission. For instance, Wang et al. reported on an organic mechanoluminescent luminescent progenitor (TPE-2-Th and TPE-3-Th), which, when combined with the presence of aggregation-induced emission, can produce very bright ML light (dark blue), even in daylight [33]. Due to its weak non-covalent bonding, the material has a relatively stable packing mode in the crystalline state, which can be easily achieved by simple heat treatment for recoverable ML emission. Based on the ML processes and the related sensing devices, the relationship between pressure and ML intensity can be recognized successfully. This kind of ML device can be a promising candidate for constructing future touch-sensing devices for communication, information storage, and healthcare.

Apart from organic materials, inorganic ZnS:Mn particles (ZMP) are another ML material. ZMP can convert mechanical stress into visible light emission in tens of milliseconds [86]. Wang et al. developed a wafer-scale flexible pressure sensor matrix (PSM) using ZnS:Mn particles (ZMP) as an intermediate mechanoluminescent material, which is sandwiched by two transparent polymer layers (Figure 4i) [46]. Piezoelectric ZnS induces a polarized charge under pressure stimulation, which promotes the detrapping of electrons, leading to energy release and excitation of Mn^2+^ ions. With the excited ions returned to the ground state, yellow visible light emits. This operational technique enables the prompt observation of visible light emission in response to local force or pressure applied to the device. By utilizing specialized picture capture and processing technology, the device enables the recording of single-point dynamic pressure signatures as well as the generation of two-dimensional (2D) pressure maps (Figure 4j). The PSM device can collect signatures more securely by recording the signer’s handwriting graphics and signature habits (Figure 4k).

The natural touch sensation process does not only include mechanical sensation but also involves thermal sensation. In addition to mimicking the mechanical sensing functionality, temperature sensing is also highly required [87]. Many cold-blooded animals, such as venomous snakes, have pit membranes that are highly sensitive and responsive to locating warm-blooded prey at a distance. Inspired by the snake pit membranes, Giacomo et al. imitated the sensing mechanism of the pit membranes using a novel artificial pectin temperature-sensing membrane [44]. The bio-inspired temperature-sensing pectin membranes exhibit superior comprehensive performance (Figure 4l,m) [44]. The main reason for the high-performance of the pectin membranes is the ingenious mimicry of TRP receptors (Figure 4n) using Ca^2+^ current regulation similar to that of pit membranes (Figure 4o). The resultant bio-inspired pectin membranes exhibit the same sensing performance as snake pit membranes (Figure 4p) over a wide temperature range. This bio-inspired temperature-sensing material can be used as an integrated layer to fabricate an artificial skin platform to improve the device’s temperature sensitivity.

## 3. Bio-Inspired Structures for the Construction of Flexible Touch Sensors

Natural selection has played a vital role in the evolution of various biological micro- and nanostructures that endow living creatures with exceptional sensing capabilities that are essential for survival in complex environments. Taking inspiration from nature, bionic structures have inspired the design of novel touch sensors and play an important role in the improvement of sensor performance. Bionic structures by mimicking the biological structures found in nature can be employed to design the whole structure of the sensors or to design the major functional parts of the sensing devices (e.g., the active sensing layer). In recent years, bio-inspired structural designs, including hierarchical structures [49,88,89], porous structures [90], cracked structures [91], interlocked structures [92], and pyramid structures [93] have been introduced into flexible touch sensors to improve their sensitivity to mechanical vibration, pressure, wind speed, sound pressure, and human pulse. In general, these structures can be divided into three categories: biomimetic plant structures, biomimetic animal structures, and bionic human structures. The unique structural properties of animal bodies and hair, plant roots, leaves, flowers, and fruit, as well as the human skin and bone body, can be imitated to fabricate with special materials to create highly sensitive flexible touch sensors (Table 2).

### 3.1. Biomimetic Plant Structures

In nature, most plants have evolved a variety of unique structures to resist the gnawing of insects and the trampling of animals. Furthermore, these structures can allow plants to receive sunlight, attract insects to disperse pollen, or help disperse seeds further afield. Inspired by these structures, researchers have developed a variety of bionic plant-inspired touch sensors with enhanced performance. For example, leaves, seeds, flowers, and pollen of plants have topological surfaces rich in biological hierarchical structures [98,99]. Highly sensitive pressure-sensing layers are suitable for using these natural structures as soft templates. Specifically, Epipremnum aureum leaf [100], Calathea zebrine leaf [51], lotus leaf [101], and rose petal [102] can be used as biological templates for the manufacture of low-cost and wearable tactile sensors.

Leaves, flowers, seeds, and pollen are the most common parts of plants. They are important organs that help plants survive and are often used as templates for fabricating sensing microstructures. For instance, to make flexible ionic skins, a dielectric layer can be created from a biomimetic microstructured ionic gel (MlG) with homogeneous cone-like surface microstructures [51]. The average surface height of MIG film is ~25 μm, and the intercone distance is 34 μm, which can effectively replace the micropyramids fabricated through complex photolithographic processes and anisotropic etching. The bio-inspired MIG film could be sandwiched by two flexible electrodes to form a capacitive touch-sensing skin. Figure 5a shows the Calathea zebrine leaves and the structures of the flexible MIG-based skin. The microstructures on the surface of the ionic gel were fabricated using a low-cost soft lithography method [103]. The Calathea zebrine leaf template was first used for the first molding, and then the ionic gel solution was cast onto the template for the second molding. An ion gel film formed after peeling off features a microcone array resembling the surface of leaves and is used as an active dielectric layer. Figure 5b shows the SEM image of the second template. From the random statistical distribution, the average diameter of the holes is approximately 35 μm. As shown in Figure 5c, uniform microcones were replicated after two molding processes. The sensing properties of the flexible capacitive sensing skin were obtained by applying specific pressure and measuring the change in capacitance, as presented in Figure 5d. Such MIG-based skin has high tactile sensitivity (54.31 kPa^−1^), exhibiting promising application in human–machine interaction and health monitoring.

The flowers of many plants have very special structures. For example, the rose petals consist of multiple hierarchical structures [104,105]. The hierarchical structures of the rose petals combined with the surface folds can be used as a natural structure template for constructing flexible touch sensors. Yu et al. create multiscale hierarchical PDMS stamps with active layers of nanowrinkled PPy using a rose petal template and a surface wrinkling pattern [94]. The PDMS stamp structures were made using a rose petal as a biotemplate, as shown in Figure 5e. Then, PPy film was applied to the duplicated PDMS stamp using oxidative polymerization, followed by self-wrinkling processing of the PPy layer. The PPy/PDMS stamps with multiscale hierarchical morphologies were used to enhance the sensing capabilities of the piezoresistive touch sensors. The biomimetic PPy/PDMS stamps shown in Figure 5f are composed of multiscale surface structures, including millimeter veins, micro-nipples, nanofolds, and nanowrinkles. The nanofolds are aligned along the direction of longitude and were uniformly distributed on the surface of every micropapilla (Figure 5g). As shown in Figure 5f,g, the nanowrinkled PPy film was formed on the nanofolds of the PDMS stamp [106]. Figure 5h shows that light irradiation can enhance the pressure response of the sensor, and the lowest detection limit can be reduced to 0.41 Pa under light irradiation. The multiscale hierarchical structure brings excellent pressure-sensing performance: a high sensitivity of 70 kPa^−1^ at 0.5 kPa, a low detection limit of 0.88 Pa, and a fast response time of 30 ms [94]. These characteristics can be attributed to the rose petal templated millimeter/micro/nano multiscale structures combined with the surface wrinkling patterns.

Plants spread their seeds in different ways, and the shape and arrangement of their seeds have inspired the design of touch sensors. Boutry and colleagues report a biomimetic electronic skin [49]. The electronic skin consists of a set of capacitors and is made of orthogonally positioned electrodes on the top and bottom of a carbon nanotube (CNT) embedded in a polyurethane (PU) matrix. An intermediate thin dielectric film was used as the electrical insulation layer of the capacitors. A pyramidal microstructure was designed as the top electrode layer of the electronic skin, featuring a foliated helical arrangement with multiple helices running clockwise and counterclockwise simultaneously, which was inspired by the head of a sunflower [107] (Figure 5i,j). These microstructures allowed the PU to bend elastically and reversibly in response to external pressure, with increased sensitivity [108]. Figure 5k displays the response characteristics of the biomimetic e-skin. Five-by-five capacitor sensor arrays are separated into orthogonal and spiral pyramid grids (Figure 5l). The picture shows that the spiral grids are significantly superior to orthogonal grids.

Inspired by the structural feature of pollen-based microcapsules, Wang et al. describe an e-skin sensor made by adding natural capsules to a composite film made of PDMS [95]. The general design idea of the e-skin device based on sunflower pollen (SFP) is shown in Figure 5m. Figure 5n,o illustrate the fabrication of the electronic skin sensor. Through functionalization and coating treatment, the treated SFP microcapsules were encapsulated into the composite matrix, forming a 3D conductive network. To examine the sensitivity of the sensor, the current changes under different static pressures were detected to calculate the specific sensitivity (Figure 5p). This biomimetic design proved to be highly efficient in pressure detection, with a high sensitivity of 56.36 kPa^−1^ in static operations.

### 3.2. Biomimetic Animal Structures

In nature, many animals live in darkness, cramped, muddy, and other extreme environments. They can perceive their environment through their unique sensing organs with special structures to sense danger and access nutrients. Their environmental adaptation strategy can be imitated to fabricate high-performance sensors with similar biomimicking structures to perceive external stimulations, e.g., touch, pressure, strain, vibration, and so on [109]. For example, fishes with lateral lines [110] and pinnipeds and rodents with hairy whiskers [111] can pick up different signals with their probe rod. In addition, the skin of some animals is composed of hierarchical structures, which can be used as soft templates to fabricate pressure-sensing microstructure. For example, shark skin is coated in small and tooth-like scales that are ribbed along the longitudinal grooves [112]. Gecko skin comprises a complex micro- and nano-hierarchical structure [113,114]. Inspired by the unique structure of these animals, unique bio-inspired sensors with intriguing properties can be developed.

For example, spiders can sense vibrations in their environment owing to slit organs with a crack-like structure [115]. In particular, the slit geometry of these organs responds well to tiny external force variations, featuring ultrahigh mechanical sensitivity [116]. Figure 6a,b provide a schematic of the spider’s slit organ. Inspired by the spider’s slit organs, Kang et al. developed a kind of highly sensitive strain sensor based on nanoscale crack junctions [54]. Figure 6c depicts a simplified drawing of the nanoscale crack sensor. Such a bio-inspired crack-structured sensor demonstrates extraordinarily high sensitivity to physiological signals and vibrations from the outside world. The crack gap of this sensor widens with strain, as seen in Figure 6d,e, and even when subjected to no strain, there exists a minuscule gap (approximately 5 nm) between the adjacent crack edges, indicating that not all the fracture edges are in total contact each other. A sensor network of 64 pixels with diameters of 5 cm × 5 cm was built (Figure 6g) to show the device’s capacity to detect minute mechanical vibrations. A piezoelectric vibrator was put on the blue-boxed region of Figure 6g as a vibration source, and a piece of PDMS was placed on the red-boxed region to impart a static pressure to the sensor matrix.

In addition to insects, mammals also have unique structures that can inspire the design of artificial touch sensors. Mice and other rodents may readily move around in the dark using their tactile system, which is made up of hairy whiskers and mechanoreceptors (Figure 6h). The leverage effect endows the mice with a sensitive tactile perception functionality [117]. Inspired by the sensing principle of mouse whiskers, An et al. designed a bendable biomimetic whisker mechanoreceptor (BWMR) for tactile perception [96]. The sensory nerves used to track the biomimetic whiskers’ deformation are two metal electrodes coated on a biomimetic hair follicle (Figure 6i). The potential equilibrium between the two electrodes is disrupted when the whisker swings because it causes a free charge transfer between them. The sign and magnitude of the transmitted charge, respectively, may be used to determine the direction and amplitude of the stimulations. The BWMR can differentiate a minuscule force of 1.129 μN thanks to the leverage effect, and the performance may be further enhanced by lengthening the whisker. The BWMR sensor’s high mechanical sensitivity makes it an excellent tool for detecting little mechanical disturbances in the surroundings (Figure 6j–l).

### 3.3. Bionic Human Structures

Similar to the creatures in nature, the human body also has unique structures or geometries, which play an extremely important role in hearing, vision, touch, smell, and taste functions. Research on somatosensory systems of the human body has provided many inspirations for the design and fabrication of touch sensors and artificial electronic skins (e-skins). Much effort has been made to mimic the structural properties and geometry characteristics of the human hand, such as fingerprint structure [118], spinosum structure [119], foot structure [42], and the hierarchy of the fingertip [97].

The epidermal–dermal junction’s intermediate ridges have been proven capable of improving the tactile perception of mechanoreceptors in human tactile systems [120]. It is known that intermediate ridges with the shape of interlocked microstructures exist between the epidermis and dermis, as shown in Figure 7a. Inspired by the interlocked epidermal–dermal ridges in human skin, Park et al. explored a kind of stretchable electronic skin with interlocked microdome structures (Figure 7a) [53]. The electronic skin was fabricated based on conductive composite elastomer films with surficial hexagonal microdome arrays. The conductive film with the microdome pattern is shown in Figure 7b. The sensor was first subjected to a normal force followed by a known shear force for testing the normal- and shear-force sensing characteristics (Figure 7c). When the normal and shear forces are applied, the surface of the microdomes in the interlocking structure is immediately deformed, which increased the contact area between the microdomes and decreased contact resistance. An electronic skin with 3 × 3 sensing pixel arrays is shown in Figure 7d. The electronic skin displayed several resolvable feedbacks when a finger was loaded onto the devices in various orientations (Figure 7d), which demonstrates the accurate perception of the direction and strength of the touch force.

Human fingers are highly sensitive to touch stimulations, where special fingerprints play an important role. As shown in Figure 7e, along with the wavy patterns in the fingerprint, the finger skin consists of layers of epidermis and dermis. The living cells that make up the dermal layer are electrically conductive, but the dead cells that make up the epidermal layer are insulating. Inspired by fingerprint anatomy, Lee and colleagues design a geometrically asymmetric TENG with paired electrodes with a microelectrode (u-electrode) on a microstructured TENG (u-TENG) [47]. The geometrically asymmetric paired electrodes mimic the wavy patterns of the outermost structure of the fingerprint. The u-TENG is made up of a conformal u-electrode and microstructured dielectric material (Figure 7f). The TENG has a high pressure sensitivity as well as dependable and durable pressure-sensing capability due to the u-electrode. In particular, the arterial pulse and acoustic vibrations can be detected by the pressure sensors based on u-TENGs. The sensors based on u-TENGs could be utilized for both voice control and vascular monitoring.

In human feet, the foot’s arches serve as a natural cushioning structure that can withstand significant impact and protect our bodies from harm. Inspired by the arch structure of the foot (Figure 7g), Song and colleagues create a wearable piezoresistive-type pressure sensor using a novel Janus graphene film (JGF) with a consistent arch-shaped convex-concave structure on both surfaces (Figure 7h) [42]. Due to the distinct arch structures, JGF-based pressure sensors have a wide detecting range since they can maintain high effective stress with little strain. The sensor’s sensitivity is shown in Figure 7i. The FGF-based (microarch-free graphene film) pressure sensor was also created to verify the mechanism, and it displayed a desirable response even at high pressure. The JGF-based pressure sensors may be utilized to monitor the pulse wave in addition to detecting excessive pressure (Figure 7i,j). As shown in Figure 7j, periodic pulse waveforms can be acquired with the JGF-based pressure sensors, which is in good consistence with the typical waveforms for the arteries [121]. In addition, the usual waveforms, such as the beginning point (S), percussion wave (P), tidal wave (T), incisura wave (C), and diastolic wave (D), can be detected and are distinguishable, as shown in Figure 7j. In the meanwhile, it is possible to compute the age index (ΔI) and the time delay (Δt) to monitor health conditions [122].

The human fingers consist of soft tissues and rigid bone. Inspired by the finger structures of humans (Figure 7k), Zhang and colleagues design a novel rigid-soft hybrid tactile sensor (RSHRS) (Figure 7l) [97]. For one sensory unit, a rigid-soft hybrid force-transmission layer is created by embedding stiff pillars into the soft elastomer matrix of the top dome-shaped layer. This structure design simulates a limb structure with bones embedded in soft muscles, which can effectively transmit external stimuli to internal structures. The sensory layer was made from piezoelectric film with five electrodes. Similar to human skin’s mechanoreceptors for detecting dynamic stimuli is this layer. The bottom layer, which supports the sensing layer, is constructed from supple material to resemble the skin dermis. As illustrated in Figure 7m, the force loading causes a bending deformation in the RSHTS piezoelectric sensing film, which generates trustworthy piezoelectric signals. The overall sensor array nonetheless has high flexibility even though the tactile sensor has a rigid-soft hybrid construction, as shown in Figure 7n. Along with adaptability, the RSHTS also showed extremely high sensitivity and a wide frequency spectrum (Figure 7o). Zhang et al. looked at the connection between the applied force and the sensor’s output charge at various frequencies. As shown in Figure 7o, for higher frequencies (200–600 Hz), the sensitivity of RSHTs reduces due to the viscoelastic properties.

## 4. Conclusions and Outlook

Biogenic materials and structures possess unique properties and characteristics, which inspired the novel design and facile construction of next-generation touch sensors. Biomaterials exhibit extraordinary biological and physicochemical properties that align well with the human body and enable sensor functionalities beyond conventional materials. For instance, hydrogel materials can be designed with various unique properties by incorporating functional groups or nano-fillers into their dynamic network. Self-repairing materials, with outstanding self-healing ability and sensing properties, are ideal for constructing artificial electronic skin. Additionally, nature-inspired materials like ion pumping-like materials, mechanical light-emitting materials, and temperature-sensing films expand the possibilities of wearable touch sensors. On the other hand, the unique structures of animals and plants in nature inspire the design of highly sensitive flexible sensors. Each organ of a plant has its own unique structure, which inspires the design of sensitive touching sensing structures. Animal skin and hair also have unique structures with excellent tactile perception capabilities. These structures can also be imitated to construct novel touch sensors with unconventional properties. These bio-inspired touch sensors by mimicking the biogenic materials and structures exhibit desirable comprehensive performance for touch-sensing, including good softness, high stretchability, self-healing capability, low detection limit, direction resolving capability, and so on. This review highlights the recent advances in using biomaterial and bionic structures to construct novel touch sensors and discusses the advantages of bio-inspired sensors in comparison with conventional touch sensors.

Moreover, the synergy of biomaterial mimicking and biological structure mimicking can further enhance the overall performance of flexible touch sensors. Wearable touch sensors that emulate both biomaterials and biological structures offer not only distinct material properties (from biomaterials) such as high toughness, mechanoluminescence, self-healing, and biocompatibility but also exhibit boosted detection sensitivity and other sensor performance (from biological structures). The combination of biomimetic materials and biomimetic structures greatly enhances the overall and comprehensive properties and capabilities of touch sensors, particularly in specialized applications. The design of biomimetic tactile sensors generally necessitates an all-sided consideration of both materials and structures. Through optimization of these two aspects, numerous wearable touch sensors with novel and exceptional performance can be designed and fabricated.

Significant progress has been made in enhancing the functionality of wearable touch sensors, with successful demonstrations of various concepts, characteristics, and applications based on bionic materials and structures. However, pushing bionic touch sensors toward practical applications faces several challenges. The first challenge lies in the full mimicking of large-number and high-density tactile sensing units in natural human skin. Human skin possesses a high density of mechanical receptors, amounting to tens or even hundreds per square centimeter [123]. Even though some touch sensors can be assembled into an array, the number and density of these sensor arrays are far from that of natural human skin. Second, the large-scale and cost-effective construction of bionic touch sensors presents another major issue. The complex material system and complicated fabrication processes greatly hinder the practical manufacturing and applications of bionic touch sensors. Addressing this challenge necessitates the development of easily available raw materials and facile fabrication technologies enabling large-scale, low-cost, and highly productive production. The third challenge could be the precise and reproducible preparation of bionic touch sensors. Touch-sensing structures derived from biological templates (e.g., leaf [51], petal [104], and so on) often exhibit random, irregular, and unrepeatable geometrical characteristics. Therefore, the prepared bionic touch sensors based on such biological templates exhibit different response performances from batch to batch, compromising the reproducibility and reliability of the bionic touch sensors. The fourth issue is related to the stability and robustness of bionic touch sensors. Flexible touch sensors based on bionic materials and structures generally lack durability and succumb to damage under harsh environments (e.g., mechanical impact, stretching deformation, excessive cycling, drying of water, long-term storage, etc.). For example, hydrogel-based touch sensors can gradually dehydrate, resulting in the variation of their touch-sensing behaviors. Despite the aforementioned challenges, biogenic materials and structures provide a wealth of theoretical and technical inspirations for flexible touch sensors. In the future, it is anticipated that an increasing number of new paradigms of bio-inspired touch sensors will be proposed and demonstrated with the aforementioned issues well resolved.

## Figures and Tables

**Figure 1 biomimetics-08-00372-f001:**
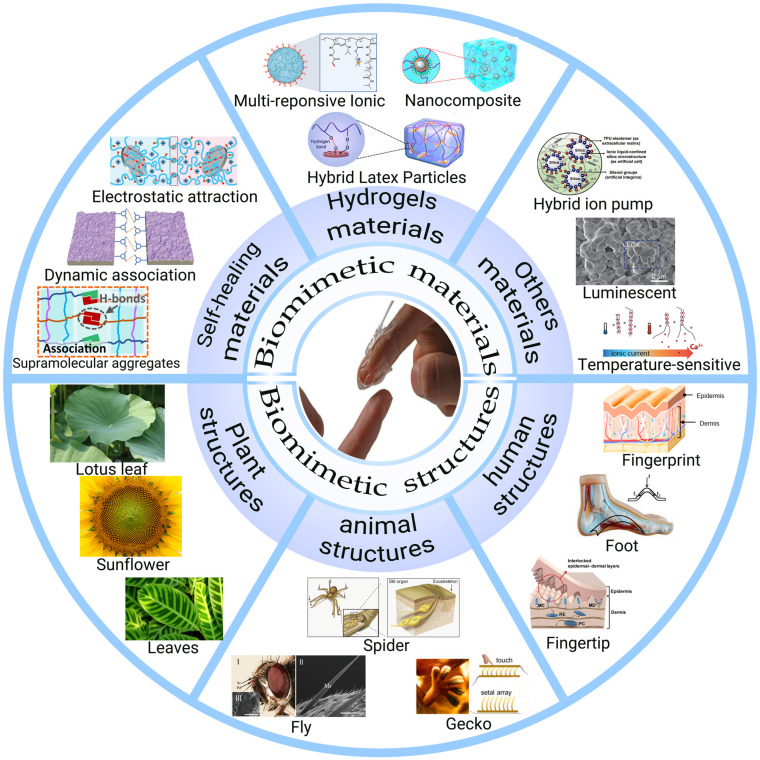
Overview of inspired wearable touch sensors by mimicking the properties of materials and structures in nature. (Reproduced with permission from Refs. [38,39,40,41,42,43,44,45,46,47,48,49,50,51,52,53,54,55], Copyright 2021, Advanced Materials Technologies; Copyright 2019, Chemistry of Materials; Copyright 2022, Nano Energy; Copyright 2019, Nat Commun; Copyright 2018, Advanced Electronic Materials; Copyright 2020, Adv Mater; Copyright 2017, Science Robotics; Copyright 2020, ACS Appl Mater Interfaces; Copyright 2015, Adv Mater; Copyright 2022, Nano Energy; Copyright 2016, Small; Copyright 2018, Science Robotics; Copyright 2022, Adv Sci; Copyright 2018, Advanced Functional Materials; Copyright 2023, Biomater Sci; Copyright 2014, ACS Nano; Copyright 2014, Nature; Copyright 2023, Nat Commun).

**Figure 2 biomimetics-08-00372-f002:**
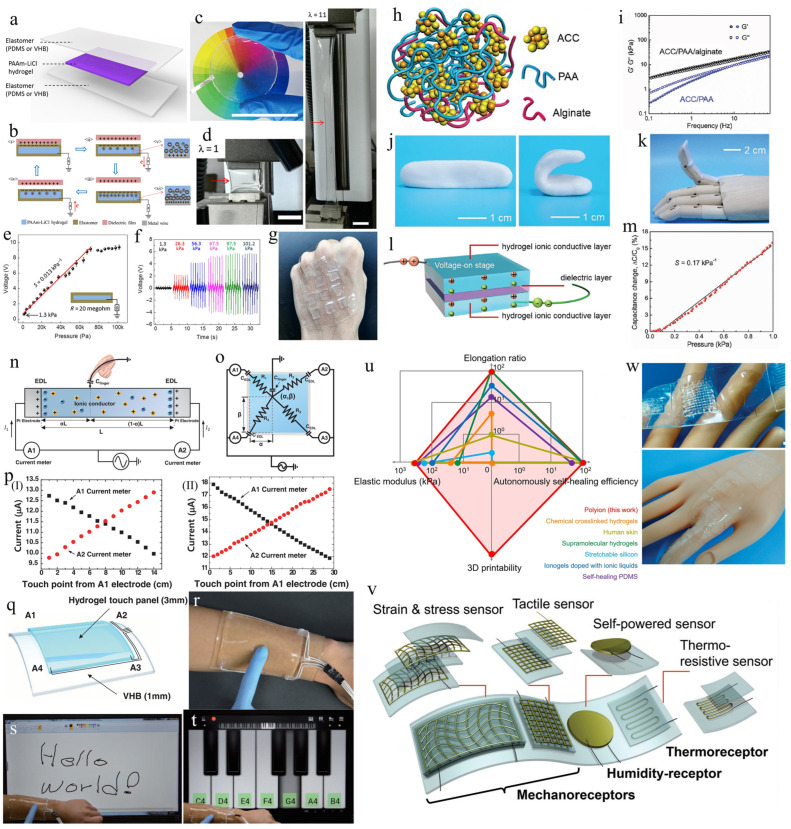
Fabrication of flexible touch sensors based on soft hydrogel materials. (**a**) Diagram of the sandwich-style STENG sensors. (**b**) Scheme of the working mechanism of the STENG sensors. (**c**) A STENG sensor that is transparent to all visible colors. (**d**) A STENG sensor at original state (ε=0) and stretched state (ε=1000%). (**e**) Variation in the voltage output across a resistor (20 megohm) at the peak amplitudes. (**f**) Representative voltage outputs of STENG sensor under five distinct pressures. (**g**) A photograph showing a hand with an affixed 3 × 3-pixel touch sensor. (Reprinted with permission from Ref. [59], Copyright 2017, Science Advances) (**h**) The chemical composition of the ACC/PAA/alginate mineral hydrogel. (**i**) Diagram showing the relationship between the frequency and the storage (G′) and loss (G″) moduli of the ACC/PAA/alginate and ACC/PAA hydrogels. (**j**,**k**) Images demonstrating the shape-change ability of the ACC/PAA/alginate hydrogel. (**l**) Schematic showing the design of the ionic skin based on ACC/PAA/alginate hydrogel. (**m**) Response curve of the hydrogel-based pressure sensor. (Reprinted with permission from Ref. [60], Copyright 2017, Adv Mater) (**n**) Schematic of a 1D ionic touch-sensing strip. (**o**) Diagram illustrating the working principle of a 2D ionic touch panel positioning system. (**p**) The linear relationship between the current and the distance of the touch point at (**I**) ε=0 and (**II**) ε=200%. (**q**) Schematic illustrating an epidermal touch panel established on a VHB substrate. (**r**) The touch panel was attached to an arm. (**s**,**t**) Human–machine interfacing applications with the epidermal touch panel, including s) writing words and (**t**) playing keyboards. (Reprinted with permission from Ref. [61], Copyright 2016, Science) (**u**) Comparison in properties of polyionic elastomer, human skin, and other reported skin-like materials. (**v**) Diagram showing the different bionic receptors on the artificial skin. (**w**) Photographs of the polyionic skin. scale bar: 1 cm. (Reprinted with permission from Ref. [62], Copyright 2017, Materials Horizons).

**Figure 3 biomimetics-08-00372-f003:**
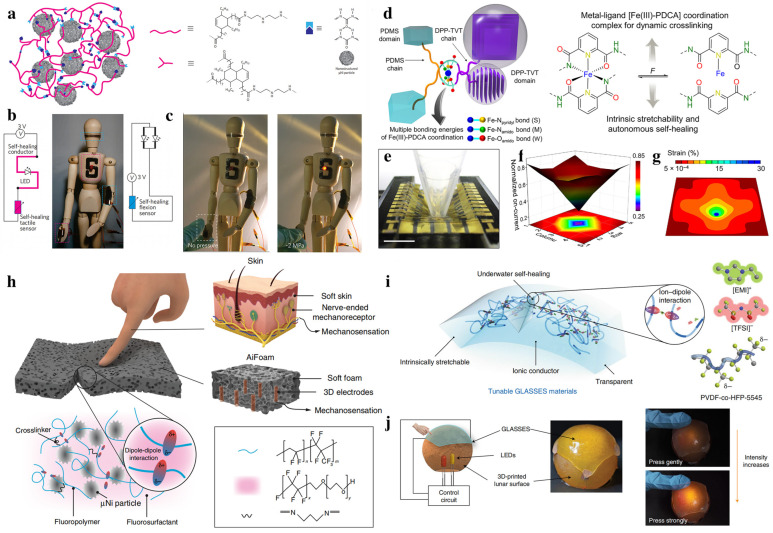
Construction of flexible touch sensors based on self-healing materials. (**a**) Schematic diagram showing the interaction between oligomer chains and the μNi particles in the developed supramolecular polymers. (**b**) Schematic circuit and picture showing the touch sensors attached to a wooden mannequin. (**c**) Pictures showing that the LED intensity can be enhanced as tactile pressure increases. (Reprinted with permission from Ref. [83], Copyright 2012, Nat Nanotechnol) (**d**) Schematic demonstrating the Fe(III)-PDCA complexation. (**e**) Photograph showing a plastic rod poking the stretched active-matrix transistor array. (**f**,**g**) Simulation result of (**f**) Normalized on-current and (**g**) strain distribution by poking the array. (Reprinted with permission from Ref. [63], Copyright 2019, Science) (**h**) Schematic illustration of the AiFoam material created by replicating the somatosensory innervation system of human skin. (Reprinted with permission from Ref. [64], Copyright 2020, Nat Commun) (**i**) Schematic illustration showing the self-healing mechanism in the tunable GLASSES material based on highly reversible ion–dipole interactions. (**j**) Schematic showing a conformable pressure sensor on a spherical “lunar” surface. (Reprinted with permission from Ref. [65], Copyright 2019, Nature Electronics).

**Figure 4 biomimetics-08-00372-f004:**
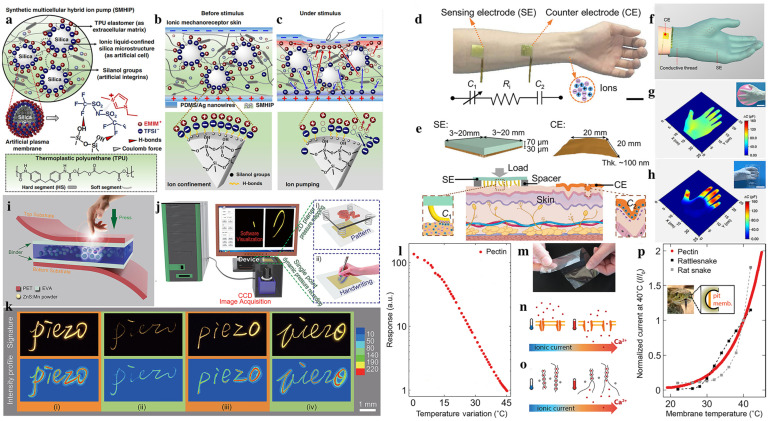
Flexible touch sensors based on other bionic materials. (**a**) Design schematic of the SMHIP made of ionic liquids ([EMIM^+^] [TFSI^−^]), silica, and TPU. (**b**,**c**) Schematic illustrating the working principle of bio-inspired ionic mechanoreceptor skin before and after applying mechanical stimulations. Inset: The amplified view of the artificial mechanoreceptor’s plasma membrane under deformation. (Reprinted with permission from Ref. [41], Copyright 2019, Nat Commun) (**d**) Photograph of the SEMS with an arm laminated with a sensing electrode (SE) and a counter electrode (CE). (**e**) Schematic of the SEMS’s layout after being affixed to the skin. (**f**) A 3D schematic illustration of the SEMS-based smart glove. (**g**,**h**) Capacitance mapping of the smart glove when the subject holds (**g**) a balloon and (**h**) a beaker. (Reprinted with permission from Ref. [66], Copyright 2021, Nat Commun) (**i**) Schematic structure of the PSM device. (**j**) Schematics showing the picture acquisition and processing system, exhibiting the single-point dynamic pressure recording and 2D planar pressure mapping method. (**k**) Demonstrations of using PSM devices to record the signing habits of four signees. (Reprinted with permission from Ref. [46], Copyright 2015, Adv Mater) (**l**) Temperature-response characteristics of the artificial skins based on pectin. (**m**) Picture showing the pectin films with good flexibility. (**n**,**o**) Molecular working mechanism of pit membrane in rattlesnake and the cross-linked pectin films. (**p**) Comparison in response behaviors of pectin film-based temperature sensors with pit membranes. (Reprinted with permission from Ref. [44], Copyright 2017, Science Robotics).

**Figure 5 biomimetics-08-00372-f005:**
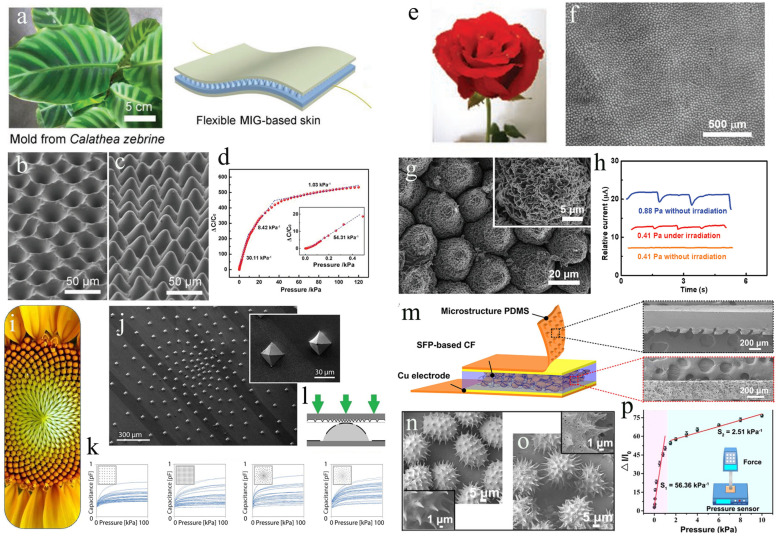
Bionic touch sensors based on the special structures of plants. (**a**) A flexible sensor made with an MlG dielectric layer. (**b**,**c**) SEM pictures of the molding templates. (**d**) Response behavior of the sensor in the pressure range of 0.1 Pa to 115 kPa. (Reprinted with permission from Ref. [51], Copyright 2018, Adv Mater) (**e**) Schematic representation of the rose petal-shaped template. (**f**,**g**) SEM images of the PPy/PDMS stamps. (**h**) The signal responses of the sensor, showing a low detection limit. (Reprinted with permission from Ref. [94], Copyright 2020, Adv Mater) (**i**) The organizational structure of sunflower. (**j**) SEM images showing the top electrode layer with a pyramid-shaped spiral grid. (**k**) Response characteristics of pyramidal orthogonal and spiral grid arrays. (**l**) Lateral view of the electronic skin with a spiral grid. (Reprinted with permission from Ref. [49], Copyright 2018, Science Robotics) (**m**) Conceptual illustration of flexible touch sensors based on an interlocking geometry with encapsulated SFP microcapsules. (**n**,**o**) SEM images of SFP microcapsules and MWCNT/SFP microcapsules, respectively. (**p**) Relative changes in the current of the sensor under different applied pressures. (Reprinted with permission from Ref. [95], Copyright 2017, Nano Energy).

**Figure 6 biomimetics-08-00372-f006:**
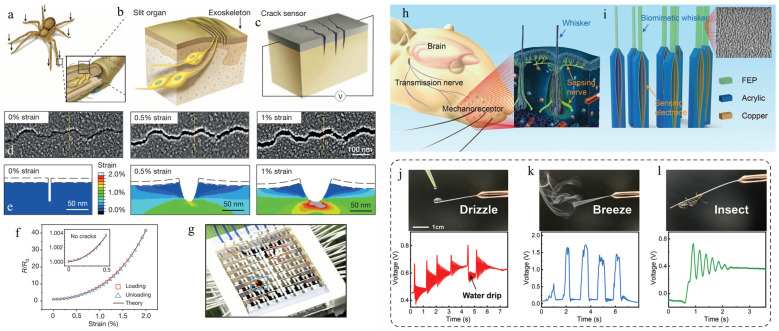
Flexible touch sensors inspired by structures of insects and animals. (**a**–**c**) Illustrations demonstrating a spider’s sensitive organs at its legs for detecting external stresses and vibrations (black arrows). The leg joint between the metatarsal and tarsal bones is the position where the sensory organs are located. (**d**,**e**) SEM images and finite-element modeling results of the zip-like crack under different strains. (**f**) Change in resistance of the sensor under different strains. (**g**) Picture showing an 8 × 8 array of the crack strain sensor. (Reprinted with permission from Ref. [54], Copyright 2014, Nature) (**h**) Schematic illustration showing a rat’s whisker sensory system and the local schematic representation of the whisker mechanoreceptor. (**i**) Schematic representation of the bionic structure of the BWMR. (**j**–**l**) Response behaviors of BWMR to weak perturbations, such as drizzle (**j**), breeze (**k**), and insect crawling (**l**). (Reprinted with permission from Ref. [96], Copyright 2021, Adv Mater).

**Figure 7 biomimetics-08-00372-f007:**
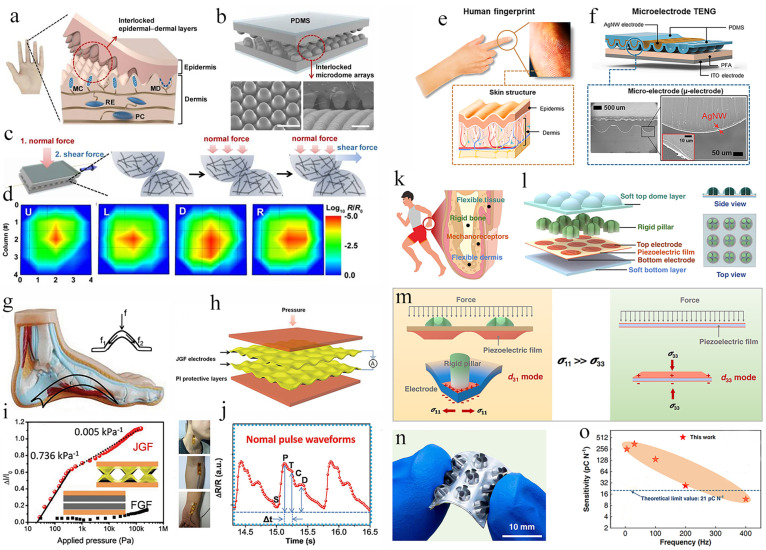
Flexible touch sensors inspired by the structures in the human body. (**a**) Schematic of human skin structure with interlocked epidermal–dermal layers. (**b**) Schematic of an interlocked microdome array used for fabricating highly sensitive touch sensors. (**c**) Schematic showing the deformation of the microdomes under stepwise application of normal and shear forces. (**d**) Schematic diagram of the electronic skin for detecting different pressing directions of the finger: left (L), right(R), up(U), down(D). (Reprinted with permission from Ref. [53], Copyright 2014, Acs Nano) (**e**) Picture and schematic illustration of human fingerprints. (**f**) Illustration and SEM images of fingerprint-inspired u-TENGs. (Reprinted with permission from Ref. [47], Copyright 2022, Nano Energy) (**g**) Anatomical model of the human foot with arch structure. (**h**) Structural scheme of the fabricated pressure sensor. (**i**) Sensitivity of JGF-based and FGF-based pressure sensors. (**j**) Characteristic peaks of pulse waveforms measured by the JGF-based pressure sensors. (Reprinted with permission from Ref. [42], Copyright 2018, Advanced Electronic Materials) (**k**) Illustration showing the anatomical structure of the fingers in the human body with different compositions. (**l**) Finger-inspired rigid-soft hybrid piezoelectric tactile sensor. (**m**) Deformation modes and working principles of the tactile sensors made from different piezoelectric sensing structures. (**n**) Photograph of an RSHTS array with 3 × 3 sensing units. (**o**) The sensitivity of the hybrid piezoelectric tactile sensor. (Reprinted with permission from Ref. [97], Copyright 2022, Nature Communications).

**Table 1 biomimetics-08-00372-t001:** Summary of wearable touch sensors based on biomimetic materials.

	Key Material	Sensitivity	Working Range	Stretchability	References
Hydrogels	PAAm-LiCl hydrogel	0.013 kPa^−1^ (TENG)	T 0–60 °C *p* > 1.3 kPa	uniaxial strain 1160%	[59]
	ACC/PAA/alginate hydrogel	0.17 kPa^−1^	*p* < 10 kPa Weight ≈ 20 mg	N/A	[60]
	PAAm hydrogel including LiCl	N/A	*p* < 100 kPa	>1000% areal strain	[61]
	Polyacrylamide/NaCl hydrogel	94% (between the output voltages and relative humidity changes)	T: 5–95 °C *p*: 0–5 kPa	0.11000% strain	[62]
Self-healing materials	DPP-TVT-PDCA	GF: 5.75 × 10^5^ with up to 100% strain	0–30% strain	Fracture strain, >1300%	[63]
	Cross-linked PVDF-HFP/fluorosurfactant foam	Resistive: 0.0982 kPa^−1^ Capacitive: 0.378 kPa^−1^	*p* < 10 kPa	230% strain	[64]
	GLASSES	N/A	N/A	2000% strain	[65]
Other materials	SMHIP	48.1–5.77 kPa^−1^	*p*:0–135 kPa	832.8% elongation	[41]
	Skin electrode	~1.3 kPa^−1^ (*p* < 3 kPa); 1.3–11.8 kPa^−1^ (*p*: 3–4 kPa); 11.8–2.8 kPa^−1^ (*p*: 4–15 kPa)	Limit of detection ~0.2 Pa	~10%	[66]
	ML materials (ZMPs)	0.7–2.2 kPa^−1^	*p*: 0.6–50 MPa	N/A	[46]
	Temperature-sensing materials (Pectin)	At least 10 mK in the 45 K range	T: 10–50 °C	N/A	[44]

**Table 2 biomimetics-08-00372-t002:** Summary of wearable touch sensors based on biomimetic structures.

Bionic Object	Key Structure	Sensitivity	Working Range	Response Time	References
Calathea zebrine leaf	Cone-like surface microstructures	54.31 kPa^−1^ (*p* < 0.5 kPa)	*p* > 0.1 Pa	29 ms	[51]
Rose petal	Multiscale hierarchical structure	70 kPa^−1^ (*p* < 0.5 kPa)	0.88 Pa–32 kPa	30 ms	[94]
Seed	Foliated helical arrangement	~0.19 kPa^−1^ (*p* < 1 kPa, at the top of the hills) ~0.1 kPa^−1^ (1–10 kPa, at the top of the hills) ~0.04 kPa^−1^ (10–20 kPa, at the top of the hills)	*p* < 100 kPa	10 s (at 20 kPa)	[49]
Pollen	Microcapsule structure	56.36 kPa^−1^ (0–1 Pa)	*p* > 1.6 Pa	0.5 s	[95]
Whisker	Whisker structure	N/A	F > 1.129 μN	Not known	[96]
Skin	Interlocked microdome structures	2.21 N^−1^ (at 7 wt.% CNTs, 65 Pa, share-force)	N/A	~18 ms	[53]
Fingerprint	Wavy patterns	2.14 VkPa^−1^ (*p* < 1 kPa)	*p*: 4 mPa–20 kPa	N/A	[47]
Foot	Arch structure	0.736 kPa−1 (reduction time = 5 min)	0–1 kPa (sensitivities = 0.736 kPa−1) 1–100 kPa (sensitivities = 0.005 kPa^−1^)	72.5 ms	[42]
Finger	Limb structure	346.5 pC N^−1^ (at 30 Hz)	F: 0.009–4.3 N	N/A	[97]

## Data Availability

No new data were created.

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
