# Peer review of "Construction of Wearable Touch Sensors by Mimicking the Properties of Materials and Structures in Nature"

_biomimetics, 2023, doi:10.3390/biomimetics8040372_

Round 1

Reviewer 1 Report

Please find my attached comments

Reviewer 2 Report

I am pleased to encounter a good review paper, which is almost ready to publish. This paper covers a plenty number of recent papers pertaining to wearable biomimetic touch sensors. The authors well classified and organized them while mentioning future challenges. I found some typos as follows.

Sec. 1. The authors may cite earlier review papers on tactile or touch sensors and clarify the novelty of their work. I understand that this may be the first review paper on wearable touch sensors inspired by mimetics; yet, this should be stated in the paper.

Fig. 2. Specifically, subfigure m is of low quality with small figures. Please, try to improve the readability.

Fig. 2. Subfigure t shows piano keyboards. But, the caption says “playing chess.”

Sec. 2.2. poly(3,6 -> poly (3, 6

Sec. 2.3, Last paragraph. Nature touch -> Natural touch?

Comment: It is nice that the authors mention the future aspects of biomimetic sensors here.

Satisfactory.

Author Response

Please see the attachment。

Reviewer 3 Report

Baojun Geng and his coauthors provide a comprehensive summary of the current research status of wearable touch sensors, with a particular focus on mimicking unique properties found in natural materials and structures. The review highlights two critical aspects of touch sensors, namely their construction using various materials and the significance of their underlying structures. The manuscript has been deemed well-written, informative, and up-to-date by the reviewer, suggesting it holds value for researchers in this field. However, to further enhance the manuscript, the reviewer recommends addressing the following minor issues:

1. Addressing Overlapping Categories of Materials:

The reviewer observed that the categories of touch sensors introduced in the review exhibit some overlap. Specifically, the authors discuss sensors made from hydrogel materials, sensors with self-healing properties, and sensors constructed from other bionic materials. While these aspects are essential for the development of touch sensors, the concepts may intertwine, leading to potential confusion for readers. To improve clarity and organization, the reviewer proposes that the authors explore different materials used in constructing touch sensors and delve into the respective bionic properties of these materials. This will help establish distinct categories and better elucidate the roles of different materials in sensor design.

2. Clarification of Material and Structure Properties:

The reviewer suggests that the authors further clarify the differentiation between material and structure properties in the context of touch sensors. As the performance of touch sensors is influenced by both the materials used and their underlying structures, it can be challenging to distinguish the individual contributions of each aspect. To address this issue, the reviewer recommends discussing the co-contribution of materials and structures to the overall performance of touch sensors in the conclusion and outlook section. This will provide readers with a clearer understanding of how these two elements work in tandem to optimize sensor functionality.

In conclusion, the manuscript authored by Baojun Geng and coauthors is highly commendable for its content, and the reviewer recommends its acceptance after addressing the aforementioned minor issues. 

Reviewer 4 Report

In this manuscript, the authors provide a detailed introduction about the wearable touch sensors. The manuscript summarized the recent advances in the construction of wearable touch sensors using biomaterial and bionic structure. The existing challenges and potential applications of the wearable touch sensors are also discussed. The manuscript is well written. This work will be of interest to the researchers in the field of electronic materials, and flexible electronic, as well as to a broad audience. Although this is an interesting study, there are a number of flaws in the manuscript. Therefore, I recommend publication after some revisions.

1.     The quality of Figure 1 is very poor. The authors should improve the figure resolution or replot the figure.

2.     In Figure 2, figure panel labels are chaotic. Figure 2 should be re-organized.  

3.     In figure 5, it would be better to show some relevant representative data panel, not only show some schematic device structures or SEM images.

Minor editing of English language required

Reviewer 5 Report

The manuscript describes the key papers in the field of biomimetic tactile sensors. The manuscript is overall written exceptionally well and includes a very good choice of highlighting the most important papers in the field.

Comments:

155-156: It is mentioned that hydrogels increase the designability. What designs or functionality or performance metrics did they achieve that were not possible before implementing hydrogels?

Writing and typos:

133: “silicon-based” should be “silicone-based”

141: “perfect adhesion” What does perfect adhesion mean?

269: “stated”; is this the correct word?

340: “fabrication optical” à “fabrication of optical”

360: “electron decapitation” Is this the right word?

483: “in dynamic and static.” It seems like this phrase needs another word, like “in dynamic and static operation.”
